# Large Animal Models of Cell-Free Cardiac Regeneration

**DOI:** 10.3390/biom10101392

**Published:** 2020-09-29

**Authors:** Andreas Spannbauer, Julia Mester-Tonczar, Denise Traxler, Nina Kastner, Katrin Zlabinger, Ena Hašimbegović, Martin Riesenhuber, Noemi Pavo, Georg Goliasch, Mariann Gyöngyösi

**Affiliations:** Cardiology, Department of Medicine II, Medical University of Vienna, 1090 Vienna, Austria; andreas.spannbauer@meduniwien.ac.at (A.S.); julia.mester-tonczar@meduniwien.ac.at (J.M.-T.); denise.traxler-weidenauer@meduniwien.ac.at (D.T.); nina.kastner@meduniwien.ac.at (N.K.); katrin.zlabinger@meduniwien.ac.at (K.Z.); n1542442@students.meduniwien.ac.at (E.H.); martin.riesenhuber@meduniwien.ac.at (M.R.); noemi.pavo@meduniwien.ac.at (N.P.); georg.goliasch@meduniwien.ac.at (G.G.)

**Keywords:** cardiac regeneration, cell-free, large animal model, porcine, microRNA, growth factor, extracellular vesicles, exosomes, cardiac reprogramming, gene therapy

## Abstract

The adult mammalian heart lacks the ability to sufficiently regenerate itself, leading to the progressive deterioration of function and heart failure after ischemic injuries such as myocardial infarction. Thus far, cell-based therapies have delivered unsatisfactory results, prompting the search for cell-free alternatives that can induce the heart to repair itself through cardiomyocyte proliferation, angiogenesis, and advantageous remodeling. Large animal models are an invaluable step toward translating basic research into clinical applications. In this review, we give an overview of the state-of-the-art in cell-free cardiac regeneration therapies that have been tested in large animal models, mainly pigs. Cell-free cardiac regeneration therapies involve stem cell secretome- and extracellular vesicles (including exosomes)-induced cardiac repair, RNA-based therapies, mainly regarding microRNAs, but also modified mRNA (modRNA) as well as other molecules including growth factors and extracellular matrix components. Various methods for the delivery of regenerative substances are used, including adenoviral vectors (AAVs), microencapsulation, and microparticles. Physical stimulation methods and direct cardiac reprogramming approaches are also discussed.

## 1. Introduction

Cardiovascular medicine has a dire need for regenerative therapies. Once damaged, the adult mammalian heart lacks the ability to repair itself and replace lost cardiomyocytes, leading to progressive loss of function. Acute myocardial infarction (AMI) is followed by the formation of a non-contractile fibrotic scar, with consequent ventricular dilation and adverse remodeling, which is a hallmark feature of heart failure with reduced ejection fraction (HFrEF) [1].

Cell-based therapies have been the focal point of regenerative medicine since the early 2000s. It was hoped that injected stem cells could engraft and replace lost cardiomyocytes at the site of injury. However, the results of large stem cell therapy trials have been mixed [2,3,4,5,6].

Consequently, research into regenerative therapies has increasingly focused on cell-free approaches. The basic premise of such therapies is that endogenous reparative mechanisms can be used to induce tissues to heal themselves. This idea is rooted in the observation of high regenerative abilities in lower vertebrates [7] and neonatal mammals such as mice [8,9,10], pigs [11,12], and even newborn humans [13,14].

Cell-free therapies carry other possible advantages over cell-based therapies, such as easier handling and storage and fewer issues of histocompatibility and immunogenicity than allogeneic stem cell therapies. However, they also carry disadvantages that are yet to be overcome, such as targeted local delivery and retention. This is especially important with potent pro-regenerative molecules, as their off-target effects could be oncogenic.

Therefore, the main challenge of cardiac regeneration therapies is to pinpoint the mechanisms responsible for tissue repair and induce them in a highly targeted manner.

The investigation of these pathways has shown important differences between small and large animal models and humans [15]. While small animal models give valuable mechanistic insights into endogenous cardiac repair mechanisms, crucial differences in physiology necessitate the intermediate translational step into large animal models before this information can be applied to humans [16].

Here, we give an overview of the current state and possible future directions of cell-free cardiac regeneration in large animal models, focusing especially on results attained within the last 5 years (Figure 1).

Effects in the text are always reported as (control vs. treatment); if no second value is reported, the value is the calculated relative difference (%) between treatment and control. Unless otherwise stated, group differences reported in the text are statistically significant (*p* < 0.05). Sample size reports in the text reflect the total number of animals enrolled in the study.

## 2. Cell-Free Cardiac Regeneration Therapies

### 2.1. Stem Cell Secretomes and Extracellular Vesicles (EVs)

In the course of investigating cell-based therapies, it was discovered that the initial assumptions of local engraftment and replacement of damaged tissue by injected stem cells were not the main driver of the observed therapeutic effects. Rather, consensus has increasingly shifted toward the “paracrine hypothesis” [17]. This hypothesis states that the secretome of stem cells modulates the resident local microenvironment toward a regenerative phenotype, which triggers myocardial repair.

As a result, extracellular vesicles (EVs) [18] and their cargo, including microRNAs (miRs), long non-coding RNAs (lncRNA), and circular RNAs (circRNAs), as well as paracrine factors such as growth hormones and cytokines, which constitute the stem cell secretome, have been investigated. In addition to native stem cell secretomes, different approaches for preconditioning the cells via serum deprivation [19], hypoxia [20], as well as radiation [21], immunological [22], and electrical [23] stimulation are being studied.

#### 2.1.1. Secretomes

The first study uses a conditioned medium of cultured mesenchymal stem cells (MSCs) porcine model of AMI that came from Timmers et al. in 2007 [24]. They showed that one intracoronary (i.c.) injection of MSC conditioned medium (MSC-CM) during reperfusion was sufficient to improve left ventricular ejection fraction (LVEF) (38.8% vs. 54%, control vs. MSC-CM) and decrease infarct size (−60%) after 4 h (*n* = 30) [24]. Interestingly, fractionation studies in a mouse model of ischemia/reperfusion (I/R) injury indicated that only the CM fraction containing products >1000 kDa (100–220 nm) provided cardioprotection. This fits well into the size range of EVs, which have since then become the main candidate for conveying stem cell paracrine effects in regeneration.

The same group published a follow-up experiment in 2011 showing that 7 days of twice daily intravenous (i.v.) infusions of MSC-CM improved several hemodynamic parameters (LVEF 33.7% vs. 49.1% control vs. MSC-CM), increased capillary density, and decreased collagen deposition three weeks after AMI, also in pigs (*n* = 22) [25].

Similar results were reported in a larger (*n* = 56) porcine trial investigating both an acute (24 h) and chronic (8 week) experimental condition using i.c. injections of endothelial progenitor cell (EPC) CM [26]. The authors reported a 37% smaller infarct size in the treatment group after 8 weeks of follow-up. This was accompanied by the increased cell-surface area within the infarction zone and improved contractility and relaxation indices. The authors were also able to show that immunoglobulin G (IgG) against IGF-1 reduced or even totally neutralized the beneficial effects of EPC-CM, leading them to posit IGF-1 as the main therapeutic molecule within EPC-CM [26].

Not only stem cell secretomes, but also the apoptotic peripheral blood mononuclear cell secretome (APOSEC) has been tested in a porcine chronic infarction model. In this study, APOSEC was injected into the border zone via a percutenous catheter 30 days after AMI, increasing cardiac index (4.40 ± 3.94 l/min/m^2^ vs. 3.07 ± 2.35 l/min/m^2^) and decreasing infarct size (−40%) after another 30 days of follow-up (*n* = 16) [27,28].

The use of peripheral blood mononuclear cells (PBMCs) is attractive because they are readily available and easy to harvest compared to stem cells. The PBMCs are preconditioned using radation to induce apoptosis, which has been shown to increase their therapeutic efficacy [29].

#### 2.1.2. EVs

EVs are vesicles of cellular origin with a lipid bilayer. They are generally classified by size and biogenesis, with exosomes (30 nm–150 nm) being of endosomal origin and microvesicles (100–1000 nm) being generated through cell membrane budding. Due to the significant size overlap between these vesicle types, recent consensus statements from the international society for extracellular vesicles (ISEV) have recommended to generally refer to EVs in the 30–200 nm range as small EVs (sEVs) [18]. EVs have come into focus as important extracellular carriers of ncRNAs, particularly miRs, and they are now thought to convey a significant portion of the paracrine effect of the stem cell secretome [30].

Cardiac progenitor cells cultured in non-adhesive conditions can form multicellular aggregates called cardiospheres, which were first described in 2004 [31] and contain cells with stem cell characteristics termed cardiosphere-derived cells (CDCs), which have been extensively studied in cell-based therapies of cardiac regeneration [32,33]. Gallet et al. used both an acute (*n =* 22) and a chronic (*n* = 13) porcine model of myocardial infarction (MI) to investigate the efficacy of CDC exosomes (CDCexo) [34]. In the acute study, intramyocardial (m.c.) but not i.c. application of CDCexo 30 min after reperfusion led to a significant difference in infarct size (−26.9%) and a preservation of LVEF after 48 h. In the chronic study, the animals either received a control vehicle or CDCexo with an endovascular intramyocardial injection catheter one month after AMI. After an additional follow-up of one month, the CDCexo group showed better LVEF (35% vs. 40%) and end-systolic volume (ESV) (56 mL/m^2^ vs. 47 mL/m^2^) compared to control [34,35]. Later work by the same group reaffirmed these results using open-chest m.c. injections of CDCexo after 30 min of reperfusion (*n* = 9) [36]. They reported that the exosomal transfer of miRs, especially miR-181b, triggers anti-inflammatory macrophage polarization, which is one of the main beneficial effects of CDCexo.

Recently, Lopez et al. [36] investigated the effects of an intrapericardial injection of CDCexo 72 h after AMI (*n* = 18) [37]. Blood sampling 24 h after treatment revealed an increase in circulating anti-inflammatory CD14+ CD163+ M2 macrophages in the treatment group, as well as an upregulation of arginase-1, which is considered an M2-differentiation marker, in the pericardial fluid. While they successfully demonstrated that a minimally invasive intrapericardial adminstration of CDCexo was safe and had a local and systemic effect on inflammatory cells, they did not find any significant differences in LVEF or infarct size 10 weeks after treatment [37].

A recent 2018 study investigated injections of MSCexo into the ischemic area 2 weeks after implantation of an ameroid constrictor, which causes the gradual development of chronic ischemia (*n* = 23) [38]. Five weeks after therapy, treated pigs showed significantly higher capillary density, cardiac output, and stroke volume compared to control [38]. In a small additional experiment, the authors also tested i.v. injections of MSCexo in four pigs, which did not have a significant effect on myocardial blood flow or cardiac function.

It is interesting to note that i.c. [24,26] and even i.v. [25] administration of MSC-CM or EPC-CM was effective in improving functional heart parameters, while i.c. [34], intrapericardial [37], and i.v. [38] administration of EVs was not. Thus far, the most effective route of administration for EVs has been m.c. injection via catheter [34] or thoracotomy [36,38]. The need for the invasive administration of EVs would represent an obvious hurdle for their clinical application. Therefore, efforts are underway to engineer EVs to specifically target the ischemic heart after systemic administration. Promising examples are the use of ischemic myocardium-targeting peptide CSTSMLKAC (IMTP) [39] or a cTnI-targeted short peptide [40] to improve the EVs affinity for ischemic myocardium, although these targeted EVs have yet to be tested in large animals.

In summary, the use of secretomes or EVs from various cellular sources offers simple, rapidly scalable production and ease of use in a clinical scenario, compared to cell-based therapies. Further developments in the selection, preconditioning, or chemical/genetic manipulation of stem cells or their EVs could increase therapeutic effects even further. Thus far, large animal models have shown that these therapeutics can be administered safely and effectively. An overview of the described studies can be found in Table 1.

### 2.2. RNA-Based Therapies

Most research into the use of RNA therapies in cardiac regeneration has focused on non-coding RNAs (ncRNAs), mainly miRs but also increasingly lncRNAs and circRNAs. The variety of ncRNA targets is vast, even in the subspecialty of cardiac regeneration [41]. However, only a fraction of the available molecules has been investigated in large animal models.

#### 2.2.1. ncRNAs

A recent study by Gabisonia et al. investigated the use of a viral vector expressing miR-199a in a porcine model of AMI [42]. Their choice of miR-199a is based on the discovery of multiple microRNAs as key regulators of cardiomyocyte proliferation, with encouraging results in rodent models [43]. After 90 min of left anterior descending (LAD) ligation followed by reperfusion, they injected adenovirus-associated vector 6 (AAV6) coding for miR-199a in the border zone of the infarction. They were able to confirm that miR-199a overexpression causes cardiomyocyte cell-cycle reentry and proliferation, which resulted in significantly lower scar size (−50%) and fibrosis, as well as improved LVEF at 4 weeks (54.6% vs. 64.86%) and beyond. However, during the extended follow-up of 8 weeks, the majority of treated animals died due to lethal arrhythmias. They theorize that this was either due to an uncontrolled proliferation of poorly differentiated cardiomyoblasts or due to overexpression of the deleterious miR-199a-5p strand from the same vector. This important study encapsulates both the promise as well as the dangers of regenerative therapies and highlights the need for the tight control of dosage and time-dependent expression of these pro-regenerative factors [42].

Contrary to overexpression and upregulation, many research groups have focused on the suppression of miRs that themselves downregulate pro-regenerative pathways. miRs can be effectively silenced by using antisense oligonucleotides, termed ‘antagomirs’, which are often chemically modified to improve their uptake and binding characteristics [44,45,46].

The miR-212/132 cluster has been identified as a critical regulator of cardiac hypertrophy and cardiomyocyte autophagy [47]. Since maladaptive cardiomyocyte hypertrophy is an important mechanism of ventricular remodeling, this makes it an attractive therapeutic target in HF treatment. A locked-nucleic-acid (LNA) modified antagomir for miR-132 (called “antimiR-132”) was developed and tested in a mouse and then a very well-powered (*n* = 135) porcine model of post-AMI HF. Three days post-AMI, animals were randomized into placebo or one of three different dose-level groups. Animals were also randomized into receiving their dose at day 3 via i.c. infusion or i.v. infusion. All treated animals received a second dose at 28 days post-AMI, which was administered intravenously. The animals were followed for 56 days, and heart function was evaluated with cMRI, which showed a significant dose-dependent improvement of LVEF, NT-proBNP, fibrosis, and cell size [48].

In 2009, miR-92a was first shown to be a potent regulator of angiogenesis in a mouse model of ischemia [49]. The authors observed that the overexpression of miR-92a led to the suppression of angiogenesis, while the use of miR-92a antagomirs enhanced it. Hinkel et al. investigated the effects of an LNA antagomir-92a administerd locally (i.c.) or systemically (i.v.) shortly before reperfusion in a porcine model of AMI (*n* = 30). Only i.c. administration resulted in a significantly smaller infarct size, which correlated with improvements in LVEF and left ventricular end-diastolic pressure (LVEDP) after 7 days. They were also able to demonstrate anti-inflammatory and anti-apoptotic effects of antagomir-92a [50]. A later work by Bellera et al. used poly-lactic-co-glycolic acid (PLGA) microspheres to deliver LNA antagomir-92a during reperfusion via i.c. infusion. They were able to show that i.c. infusion of these microspheres was safe and led to high retention in the reperfused myocardium, without causing local obstruction of blood flow. After one month, they found that this single i.c. injection of microspheres resulted in significantly improved septoapical wall motion and less adverse remodeling [51].

#### 2.2.2. Coding RNAs

Another possible approach for the transient upregulation of pro-regenerative factors and pathways is the use of modified or synthetic coding RNAs, which can be optimized regarding immunological and expression characteristics. The technique was first established in 2011, where a chemically modified mRNA (modRNA) coding for murine erythropoietin was effective in raising the hematocrit of mice [52]. In the context of myocardial ischemia, it was shown that modRNA for vascular endothelial growth factor A (VEGF-A) exerted pro-regenerative effects in a mouse model [53]. Turnbull et al. demonstrated that lipoid nanoparticles carrying modRNA are also effectively transcribed in pigs, although at this time, there are no studies investigating cardiac regeneration in large animals with this method [54].

RNA therapeutics have been shown to be very potent regulators of cellular regeneration. However, due to their abundance and pleiotropic, tissue-dependent [55] functions, their side effects are hard to predict. Well-designed large animal studies have recently impressively showcased both their promise [42,48] and their dangers [42]. Other advancements are lipid nanoparticle formulations, which have been shown to increase the cellular uptake of RNA therapeutics [56]. The reservoir of possible targets waiting to be translated to large animal models is deep, offering many research opportunities in the coming decade [41]. RNA-based therapies are summarized in Table 2.

### 2.3. Growth Factors and Single Molecules

Aside from secretome, EV-, and RNA-based approaches, targets such as growth factors, proteins, as well as other molecules have been identified and investigated in large animal models.

While some of the following studies include data from small animal experiments, we will focus on results generated with respect to large mammals.

#### 2.3.1. Growth Factors

A common method for the upregulation of specific growth factors is gene therapy using vectors, such as adeno-associated viruses (AAVs) or plasmids. AAVs are attractive because of their comparatively low likelihood to integrate into the host cell genome; rather, they persist in an extrachromosomal form and are stably expressed [57]. AAV1, AAV6, and especially AAV9 are most promising for their ability to preferentially accumulate in cardiac tissue [57]. However, it must be noted that even with these improvements, viral vectors are still considered to have an unfavorable safety profile.

Vascular endothelial growth factor (VEGF) is an intuitively attractive candidate for cardiac regeneration because it promotes angiogenesis. To increase its efficacy, it is often combined with other growth factors. Examples include the use of VEGF and angiopoietin-1 (Ang1) in a porcine model of AMI (*n* = 24) [58]. The treament was injected shortly after the initiation of AMI, and the animals were followed for 8 weeks. The authors reported a robust improvement in cardiac function and myocardial perfusion with this combined treatment.

In a porcine model of chronic ischemia, an AAV carrying VEGF-A was administered with and without platelet-derived growth factor B (PDGF-B) (*n* = 27) [59]. Interestingly, after a follow-up of 56 days, only the combination of VEGF-A and PDGF-B resulted in a significant improvement of collateralization, myocardial blood flow reserve, and LVEF, while VEGF-A alone had no significant effects.

In an effort to avoid viral vectors, Bulysheva et al. assessed the use of plasmids encoding VEGF-A in a porcine model of AMI. After surgical ligation of the LAD, the plasmids were injected into the ischemic border zone, and then gene electrotransfer (GET) was performed. This led to an increase in vessel formation in the treatment group but without significant improvements in LVEF and other echocardiographic parameters. Importantly, this study demonstrates the safety of the GET approach for transient local gene therapy in a porcine model [60].

In contrast to VEGF-A, VEGF-B is less pro-angiogenic than it is cytoprotective and anti-apoptotic. Therefore, it was investigated as a potential treatment for non-ischemic dilated cardiomyopathy, using a canine tachypacing model (*n* = 53) [61]. Several different AAV vectors carrying VEGF-B167 were investigated as well as an immediate and a delayed treatment protocol. After 28 days of pacing, VEGF-B167 treated dogs exhibited significantly better left ventricular end-diastolic pressure (LVEDP), dP/dt_max_, and LVEF than control animals. The effect size was surprisingly large; LVEDP, a marker of central congestion and diastolic dysfunction, remained almost at physiological pressures (6–10 mmHg), while untreated animals developed severe heart failure with more than twice that pressure. On a cellular level, significantly less apoptosis was observed in the treatment group. These results are very encouraging, since effective treatments for non-ischemic dilated cardiomyopathy are scarce [61].

Fan et al. identified acidic fibroblast growth factor (FGF-1) and CHIR99021 as synergistic enhancers of cardiomyocyte cell cycle activity [62]. CHIR99021 is an aminopyrimidine derivative that functions as a Wnt signaling activator, which has also been shown to be able to reprogram fibroblasts to a cardiomyocyte phenotype [63]. In their study, both molecules were loaded onto PLGA nanoparticles and injected intramyocardially 15 min after reperfusion of AMI in a pig model (*n* = 12). This resulted in significantly reduced scarring as well as increased LVEF and decreased left ventricular end-diastolic volume (LVEDV) compared to control animals after 28 days of follow-up [62].

In a recent 2020 study, an i.c. injection of microencapsulated insulin like growth factor 1 (IGF-1) was administered 48 h after reperfusion of AMI in a porcine model (*n* = 24) [64]. After 10 weeks of follow-up, the treatment group exhibited significantly higher LVEF (+18%) and vascular density as well as lower collagen volume fraction, indicating decreased fibrosis.

Lastly, another 2020 study demonstrated that a 7-day i.v. infusion of recombinant human platelet-derived growth factor-AB (rhPDGF-AB) in a porcine model of AMI improved survival by 40% and increased LVEF 11.5% compared to control after one month of follow-up (*n* = 36). They were also able to demonstrate enhanced angiogenesis and increased scar anisotropy (fiber alignment), which translated to a reduction of inducible ventricular tachycardia [65].

#### 2.3.2. Other Molecules

A crucial component of left ventricular remodeling and cardiac scar formation is the extracellular matrix (ECM) [66]. An important class of proteolytic enzymes responsible for ECM turnover and remodeling are matrix metalloproteinases (MMPs), which have been shown to be dysregulated following ischemic injury [67]. It has also been demonstrated that the selective inhibition of MMPs can have a favorable effect on post-ischemic remodeling [68]. Tissue inhibitors of metalloproteinases (TIMPs) are physiological inhibitors of MMPs. Of these, TIMP-3 has been intensively investigated in mice transgenic models of ischemic injury [69]. Its therapeutic potential was recently demonstrated in a porcine model of AMI, where a single intracoronary injection of TIMP-3 shortly before reperfusion reduced infarct size (−45%) and LV dilation (−40%), and it attenuated the deterioration of LVEF after 28 days of follow-up (*n* = 17) [70].

A very recent 2020 study tested a fragment of the ECM protein agrin in a porcine model of AMI (*n* = 19) [71]. rhAgrin has previously been shown to have a reparative effect in mice [72]. First, the porcine study established that antegrade i.c. infusion was the most effective, leading to a higher rhAgrin concentration in the infarct and border zone compared to retrograde infusion into the anterior interventricular vein or m.c. injection into the border zone. Importantly, with antegrade i.c. infusion, no rhAgrin was detected in peripheral organs. Next, the authors tested whether a single i.c. infusion after reperfusion was sufficient or if a second infusion 3 days post-AMI increased the therapeutic effect. Finally, they demonstrated that a single i.c. injection of rhAgrin during reperfusion was sufficient to significantly improve LVEF and LVEDP after 3 and 28 days of follow-up. Additionally, scar size and remodeling was also positively affected by rhAgrin treatment. The mechanisms of action of rhAgrin are pleiotropic, involving multiple cell types, improving angiogenesis, reducing inflammation, and increasing cell-cycle re-entry [71].

An instructive example of the importance of location and mechanism of delivery is a 2015 study by Wei et al. [73]. They identified follistatin-like 1 (Fstl1) as a potent stimulating factor for cardiac regeneration as long as it was expressed epicardially. Following infarction, Fstl1 expression decreases epicardially but increases myocardially, which was shown not to contribute to cardiac regeneration. However, when applied via an epicardial patch, a potent regenerative response was documented [73]. This echoes the findings in zebrafish, where the epicardial expression of vascular endothelial growth factor A-A (vegfaa) increased cardiomyogenesis, but the global ectopic overexpression actually decreased repair at the injured site [74].

Ferraro et al. first established the pro-regenerative characteristics of annexin A1 (AnxA1) in a mouse model and then used a cardiotropic AAV vector to overexpress AnxA1 in pigs before inducing AMI [75]. They observed a macrophage dependent increase in VEGF-A release that resulted in increased neoangiogenesis and cardiac repair both in mice and pigs [75].

Shapiro et al. used an AAV expressing cyclin A2 (Ccna2), which was injected intramyocardially one week after AMI in pigs. They showed an improvement in LVEF of 22% relative to control and an increase in CM cell-cycle activity six weeks after treatment [76].

Another porcine model injecting a recombinant adenovirus encoding stem cell factor (SCF) 1 week after AMI found improved LVEF (+12%), decreased apoptosis, and increased capillary density after three months (*n* = 22) [77].

In an example of a porcine model of chronic ischemia, Ziegler et al. used AAVs coding for thymosin β4 (Tβ4) and its downstream signaling molecule myocardin-related transcription factor A (MRTF-A), which were either designed to be constitutively expressed or inducible via a tetracycline-sensitive promotor [78]. After implanting a reduction stent in the proximal RCA, they waited 28 days for chronic myocardial ischemia to develop. Then, they infused the AAVs into the great cardiac vein. After another 28 days during which Tβ4 and MRTF-A were either constitutively expressed or pulsed for 3 periods of 5 days, they were able to show comparable improvements between both groups and large beneficial effects compared to control, such as increased capillary density, collateralization, as well as improved hemodynamic parameters such as LVEF and LVEDP [78].

In summary, growth factors, single molecules, and combined therapies have shown great promise in recent years. Especially, the continued development of advanced delivery systems such as microencapsulation, microparticles, nanoparticles, vesicles, improved AAVs, and other vectors that are compatible with i.c. or even i.v. administration is encouraging for future clinical translation. An overview of the described studies can be found in Table 3.

### 2.4. Physical Stimulation of Regeneration

Another set of strategies relies on the use of physical forces such as shock waves, laser impulses, or electrical stimulation to trigger regenerative mechanisms. For an excellent in-depth review on the topic including small animal and in vitro experiments, see Facchin et al. [79].

#### 2.4.1. Shock Wave Therapy (SWT)

SWT uses acoustic shock waves to stimulate tissues. Nishida et al. demonstrated in 2004 that extracorporeal SWT with about 10% of the kinetic energy used for lithotripsy had beneficial effects in the setting of a porcine chronic ischemia model (*n* = 16) [80]. Four weeks after implantation of an ameroid constrictor, they applied SWT (0.09 mJ/mm^2^, 200 shots/spot) to nine ischemic spots on the heart, which was guided by echocardiography, three times within one week. After a follow-up of 4 weeks, they observed a recovery of LVEF (51 ± 2% to 62 ± 2%), increase in wall thickening fraction (WTF) (13 ± 3% to 30 ± 3%), and increase in myocardial blood flow in the treatment group, which was not observed in the control animals. Since then, multiple groups have replicated these results, although mostly with an invasive methodology that applies the SWT directly to the affected myocardial area. This invasive protocol has been tested in mice [81], rats [82], and finally in pigs [83], with very similar outcomes in the treatment group as observed by Nishida et al. The therapeutic effect seems to be driven mainly by increased angiogenesis via the upregulation of VEGF and chemoattraction of endothelial cells via stromal cell-derived factor 1 [81].

#### 2.4.2. Low-Level Laser Therapy (LLLT)

In LLLT, low-power laser impulses are used to stimulate regeneration. A 2016 study by Blatt et al. investigated the effect of LLLT on the bone marrow after AMI. At 30 min as well as 2- and 7-days post-AMI, they applied LLLT to the tibia and the iliac bone with an 808 nm laser equipped with a rigid fiberoptic, which they inserted through small skin incisions. The authors reported an increase in circulating c-kit+ cells as well as decreased infarct size (−68%) and increased regional perfusion 90 days after infarction (*n* = 12) [84].

Earlier studies published in 2001 have also demonstrated the beneficial effects of LLLT when directly applied to the infarcted area in dogs, achieving infarct size reductions of around 50% in both trials (*n* = 22) and (*n* = 50) [85,86].

#### 2.4.3. Localized High-Frequency Electrical Stimulation (LHFS)

In a pig model, localized high-frequency electrical stimulation (LHFS) (240 bpm, 0.8 V and 0.05 ms pulses) was applied to the infarcted area using a pacemaker. The stimulation started 7 days after AMI and was continued for at least 21 days. This resulted in significant differences in increase of EDV (32% vs. 12%) and pulmonary capillary wedge pressure (PCWP) (+62% vs. −17%) in the untreated and treated cohorts, respectively (*n* = 11) [87]. The authors reported no difference in scar size between the groups. Together, these results suggest that LHFS has a beneficial effect on adverse remodeling without affecting scar size.

#### 2.4.4. Bioelectrical Stimulation

The importance of cholinergic innervation in cardiac regeneration has already been demonstrated in zebrafish and neonatal mice, where pharmacological or surgical denervation causes an abrogation of myocardial repair [88]. Mahmoud et al. demonstrated that the administration of neuregulin-1 (Nrg1) and nerve growth factor (Ngf) partially rescued myocardial repair after denervation [88].

In a pig model of AMI, continuous and intermittent spinal cord stimulation (SCS) significantly increased LVEF after 10 weeks and stimulated diffuse sympathetic nerve sprouting in and around the infarcted region compared to control (*n* = 30) [89].

Another application of bioelectrical stimulation in cardiac regeneration is vagus nerve stimulation (VNS). In a crossover study using a canine model of chronic HF, 3 months of low-level VNS significantly improved LVEF, end-systolic volume, and N-terminal prohormone of brain natriuretic peptide (proBNP) (*n* = 26) [90]. It has been proposed that the major driver behind the pro-regenerative effects of cholinergic nerves lies in their interaction with the immune system, namely the polarization of macrophages to an anti-inflammatory M2 phenotype [91].

Strategies using physical forces such as shockwaves, lasers, or electrical stimulation are attractive because they can be used in a highly targeted manner with few reported systemic side effects. However, many of the current treatment protocols require invasive surgery or implantation of stimulation devices. This, combined with need for specialized equipment and expertise, has been a hinderance to their adoption in clinical practice. Future research should focus on unraveling the molecular mechanisms that give rise to the therapeutic effects of physical stimulation therapies. Then, such information could guide the further development of these methods or lead to non-invasive therapies that mimic these mechanisms. An overview of physical stimulation approaches to cell-free cardiac regeneration can be found in Table 4.

### 2.5. Direct Cardiac Reprogramming

A promising future direction for cell-free cardiac regeneration in large animal models is the concept of in situ direct cardiac reprogramming.

In contrast to the creation of cardiomyocytes from induced pluripotent stem cells (iPSCs), which requires the dedifferentiation of cells into a pluripotent state followed by cardiomyocyte differentiation, direct cardiac reprogramming directly transforms cells from their original type (usually cardiac fibroblasts) to induced cardiomyocytes (iCMs) without a pluripotent intermediary stage. This method presents several inherent advantages over iPSCs. Firstly, iPSCs can suffer from tumorgenicity, immunogenicity, and owing to their pluripotency, a higher likelihood of differentiating into unwanted cell lineages. Secondly, the main target of cardiac regeneration therapies, ventricular scar tissue, is mainly comprised of cardiac fibroblasts, which are the main substrate for direct cardiac reprogramming [92].

In 2010, Ieda et al. published the first report of direct reprogramming of cardiac fibroblasts into functional cardiomyocytes in vitro via the combination of three transcription factors, namely Gata4, Mef2c and Tbx5 (collectively abbreviated GMT) [93].

Since then, multiple experiments have been carried out in vitro and in small mammals [94,95,96]. A crucial advance was the discovery of a combination of miRNAs (miR-1, miR-133, miR-208, and miR-499) that could also be used for the direct cardiac reprogramming of cardiac fibroblasts in vitro and in vivo [97]. Of equal importance, a chemical cocktail called CRFVPT has also been discovered to be able to induce direct cardiac reprogramming in murine fibroblasts [98]. These approaches obviate the need for viral vectors, which improves their safety profile.

Thus far, no in vivo large animal models of direct cardiac reprogramming have been published. However, the method has been demonstrated in in vitro cultures of pig [99], canine [100], and human cells [99,101,102]. This, combined with the scarcity of in vivo studies of in situ direct cardiac reprogramming in chronic ischemic models, where the fibrotic scar would constitute the perfect target for this method, represents a major future research opportunity in large animal models of cell-free cardiac regeneration therapies.

## 3. Conclusions

This review gives an overview of the diverse field of cell-free cardiac regeneration with a focus on large animal models with importance for translation into clinical practice.

Cell-free cardiac regeneration is a rapidly developing area of research with a staggering variety of treatment approaches and therapeutic molecules, each with their own advantages and disadvantages that need to be overcome. A common theme among all of them is the need for minimally invasive, targeted delivery. As regenerative therapies become more potent, the precise spatiotemporal control of delivery will become an important bottleneck in their clinical application. Overall, the recent results from large animal models presented here are very encouraging, and there is no shortage of candidate molecules waiting to be translated from small animal models.

In conclusion, cell-free cardiac regeneration therapy holds the promise to fundamentally alter the current medical landscape by turning ischemic myocardial injury into a transitory rather than a progressive disease process and large animal models are a key tool for realizing this potential.

## Figures and Tables

**Figure 1 biomolecules-10-01392-f001:**
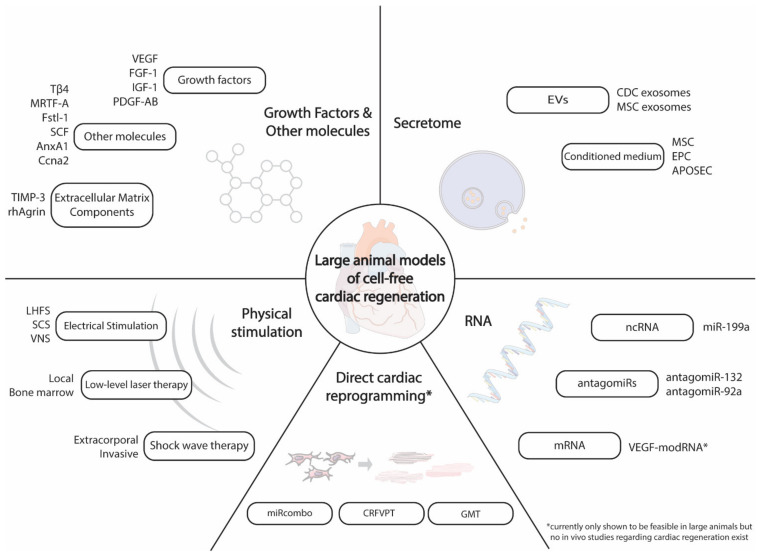
Overview of large animal models of cell-free cardiac regeneration.

**Table 1 biomolecules-10-01392-t001:** Stem cell secretomes and extracellular vesicles (EVs).

Stem Cell Secretomes and EVs.				
Therapy Delivery Method	Animal Model Sample Size Follow-Up	Main Effects	Proposed Mechanism	
MSC-CM i.v. + i.c.	Pig (AMI) (n = 30) 4 h	↑ LVEF (+32%) ↓ Infarct size (−60%)	↓ TGF-β signaling ↓ Apoptosis	[24]
MSC-CM i.v.	Pig (AMI) (n = 22) 3 w	↑ LVEF (+37%) ↑ Capillary density (+50%)		[25]
EPC-CM i.c.	Pig (AMI) (n = 56) 24 h/8 w	↑ ± dp/dt ↓ Infarct size (−37%)	↑ IGF-1 signaling	[26]
APOSEC m.c.	Pig (AMI) (n = 16) 60 d	↑ Cardiac index ↓ Infarct size (−42%)	↑ Angiogenesis ↓ Apoptosis	[27,28]
CDCexo i.c./m.c.	Pig (AMI) (n = 22) 48 h (n = 13) 60 d	↑ LVEF (+21.27%) 48 h ↓ Infarct size (−26.9%) 48 h ↑ LVEF (+13.3%) 60 d ↓ ESV (−17.47%) 60 d	↑ Anti-inflammatory macrophage polarization ↓ Fibrosis ↑ Angiogenesis	[34,35]
CDCexo m.c.	Pig (AMI) (n = 9) 48 h	↑ LVEF (+18%) ↓ MVO ↓ CD68+ macrophages		[36]
CDCexo intrapericardial	Pig (AMI) (n = 18) 10 w	↑ M2 macrophages		[22,37]
MSCexom.c.	Pig (ameroid constrictor)(n = 23)7 w	↑ Stroke volume (+33.7%)↑ Capillary density	↑ Angiogenesis	[38]

Main effects are always reported as (control vs. treatment), if no second value is reported, the value is the calculated relative difference between treatment and control. Only significant values are reported in the main effects column (*p* < 0.05). Sample size reports the total number of animals enrolled in the study. Abbreviations: m.c.: intramyocardial injection, MVO: microvascular obstruction. ↓: downregulation, ↑: upregulation.

**Table 2 biomolecules-10-01392-t002:** RNA-based therapies in large animal models of cell-free cardiac regeneration.

RNA-Based Therapies				
Molecule (Delivery Method)	Animal Model Sample Size Follow-Up	Main Effects	Proposed Mechanism	
miR-199a AAV6 mc.	Pig (AMI) (n = 19) 8 w	↑ LVEF (+17.1%) ↓ Scar size/mass (−50%) ↓ Fibrosis	CM cell-cycle reentry	[42]
antimiR-132 i.c./i.v.	Pig (AMI) (n = 156) 56 d	↑ LVEF ↓ Scar size ↓ Fibrosis ↓ CM size	↑ Foxo3 (anti-fibrotic)↑ Serca2a	[48]
antagomiR-92a i.v./i.c.	Pig (AMI) (n = 30) 7 d	↑ LVEF ↑ Capillary density	↑ Angiogenesis ↓ Inflammation ↓ CM apoptosis	[50]
PLGA antagomir-92a i.c.	Pig (AMI) (n = 27) 10 d	↑ LVEF ↓ Adverse remodeling		[51]

Main effects are always reported as (control vs. treatment), if no second value is reported, the value is the calculated relative difference between control and treatment. Only significant values are reported in the main effects column (*p* < 0.05). Sample size reports the total number of animals enrolled in the study. Abbreviations: CM: cardiomyocyte.

**Table 3 biomolecules-10-01392-t003:** Overview of the various large animal models of cardiac regeneration using growth factors, proteins, and other regenerative molecules.

Growth Factors, Proteins, and Other Molecules				
Molecule Delivery Method	Animal Model Sample Size Follow-Up	Main Effects	Proposed Mechanism	
VEGF + angiopoietin-1 AAV mc.	Pig (AMI) (n = 24) 8 w	↑ LVEF ↑ Capillary density	↑ Angiogenesis ↑ CM proliferation ↓ Apoptosis	[58]
VEGF + PDGF-B AAV i.c.	Pig (reduction stent) (n = 27) 56 d	↑ LVEF ↑ Collateralization		[59]
VEGF-A plasmid + GET mc.	Pig (AMI) (n = 37) 7 w	↑ Angiogenesis		[60]
VEGF-B167 AAV i.c.	Canine (dilated CMP) (n = 53)	↑ LVEF ↓ LVEDP ↑ dP/dt_max_	↓ Apoptosis	[61]
FGF-1 + CHIR99021 NPs mc.	Pig (AMI) (n = 12) 28 d	↑ LVEF ↑ Angiogenesis	CM cell cycle reentry	[62]
Microencapsulated IGF-1 i.c.	Pig (AMI) (n = 24) 10 w	↑ LVEF (+18%) ↓ CVF ↓ ESVi	↑ Angiogenesis ↓ Apoptosis	[64]
rhPDGF-AB i.v.	Pig (AMI) (n = 36) 28 d	↑ Survival ↑ LVEF ↑ Scar anisotropy ↓ VT	↑ Angiogenesis Fibroblast modulation	[65]
TIMP-3 i.c.	Pig (AMI) (n = 17) 28 d	↓ Infarct size (−45%) ↓ LV dilation (−40%)	MMP inhibition	[70]
rhAgrin i.c.	Pig (AMI) (n = 19) 28 d	↑ LVEF ↓ Scar size ↓ Adverse remodeling	CM cell-cycle reentry ↑ Angiogenesis ↓ Inflammation	[71]
Fstl-1 Epicardial patch	Pig (AMI) (n = 6) 5 w	↑ LVEF ↓ Scar size ↑ CM proliferation ↑ Arteriogenesis	↑ Cardiogenesis ↓ Apoptosis	[73]
AnxA1 cardiotropic AAV i.v.	Pig (AMI) (n = 7) 7 d	↑ VEGF-A Macrophage polarization	Macrophage modulation	[75]
Ccna2 AAV mc.	Pig (AMI) (n = 27) 6 w	↑ LVEF ↑ CM proliferation	CM cell-cycle reentry	[76]
SCF mc.	Pig (AMI) (n = 22) 3 m	↑ LVEF (+12%) ↑ Angiogenesis	↑ Angiogenesis ↓ Apoptosis	[77]
Tβ4 + MRTF-A AAV i.v.	Pig (reduction stent) (n = 20) 56 d	↑ LVEF ↓ LVEDP ↑ Collateralization	↓ Apoptosis ↑ Collateralization	[78]

Main effects are always reported as (control vs. treatment); if no second value is reported, the value is the calculated relative difference between treatment and control. Only significant values are reported in the main effects column (*p* < 0.05). Sample size reports the total number of animals enrolled in the study. Abbreviations: NPs: nanoparticles, CVF: collagen volume fraction, ESVi: end systolic volume (indexed to Body Surface Area), VT: ventricular tachycardia.

**Table 4 biomolecules-10-01392-t004:** Overview of large animal models of physical stimulation for cell-free cardiac regeneration.

Physical Stimulation				
Therapy Delivery Method Target Area	Animal Model Sample Size Follow-Up	Main Effects	Proposed Mechanism	
SWT Extracorporeal Ischemic myocardium	Pig (Ameroid constrictor) (n = 16) 8 w	↑ LVEF ↑ WTF ↑ Angiogenesis	↑ VEGF	[80]
SWT Invasive Ischemic myocardium	Pig (AMI) (n = 11) 6 w	↑ LVEF ↑ Angiogenesis		[83]
LLLT Invasive Bone marrow	Pig (AMI) (n = 12) 90 d	↓ Infarct size (−68%) ↑ Angiogenesis	↑ VEGF ↑ Stem cell proliferation	[84]
LLLT Invasive Ischemic myocardium	Canine (AMI) (n = 22) 14 d	↓ Infarct size (−49%)		[85]
LLLT Invasive Ischemic myocardium	Canine (AMI) (n = 50) 6 w	↓ Infarct size (−52%)		[86]
LHFS Invasive Ischemic myocardium	Pig (AMI) (n = 11) 28 d	EDV (32% vs. 12%) PCWP (+62% vs. −17%)	↓ Adverse remodeling	[87]
SCS Intermittent/continuous T1-T3	Pig (AMI + Pacing) (n = 30) 10 w	↑ LVEF ↑ +dP/dt	↑ Sympathetic nerve sprouting	[89]
VNS Implanted Electrode Vagus nerve	Canine (Microembolization) (n = 26) 6 m	↑ LVEF ↓ ESV	↓ Inflammation Macrophage modulation	[90]

Main effects are always reported as (control vs. treatment); if no second value is reported, the value is the calculated relative difference between treatment and control. Only significant values are reported in the main effects column (*p* < 0.05). Sample size reports the total number of animals enrolled in the study. Abbreviations: SWT: Shock wave therapy, LLLT: Low-level laser therapy, LHFS: Localized high-frequency electrical stimulation, SCS: Spinal cord stimulation, VNS: Vagus nerve stimulation.

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
