# Peer review of "Large Animal Models of Cell-Free Cardiac Regeneration"

_biomolecules, 2020, doi:10.3390/biom10101392_

Round 1

Reviewer 1 Report

Spannbauer et al. present an interesting, comprehensive/compact, and well-written summary of recent progress in the field of cell-free cardiac regeneration. Obviously, the entire field of cardiac regeneration has become very large and multi-facetted, therefore it makes a lot of sense to focus on the area of cell-free approaches, since  these therapies and their scientific base is anyway a more modern and promising approach in regenerative medicine than (blind) injection of stem cells. Also, the review exclusively looks at large animal models (mostly pig) as those projects usually are advanced and have most translational potential. Taken together, I like the concept of the review and have to further requests for a revision.

Author Response

Dear Reviewer 1,

thank you for your kind comments and thank you for taking the time to review our article.

Reviewer 2 Report

This manuscript is very well written and contains a good overview of cell-free cardiac regeneration therapies strategies that tested in the most clinically relevant: large animal model, pigs. The review is very important for the development of novel strategies in heart function repair after ischemic injuries like myocardial infarction.   I would like to make a suggestion to add a paragraph discussing advantages and potential risks of cell-based delivery versus cell-free therapies. 

Author Response

Dear Reviewer 2,

thank you for your suggestion. We have added a paragraph discussing advantages and potential risks of cell-free therapies in the introduction section. Thank you for taking the time to review our article.

Reviewer 3 Report

1) These authors attempted to summarize the various approaches that have been employed to regenerate heart tissue in large animals models. However, the article provides rather brief and cursory summaries of these approaches, which are each rather short and inadequate. Such brevity reduces the overall usefulness of this article for readers. There have been numerous reviews on cardiac regeneration and this overly broad review does not really add something to the existing literature in the current form.

2) The paragraph on “direct cardiac reprogramming” could be slightly off-topic since it doesn’t describe any report on large animals.

3) Each section needs a small table or a figure to summarize or illustrate what is said in the text. Figure 1 is unclear and chaotic. It should be reworked by a professional graphic designer.

4) Conclusion section is redundant and not well organized. I strongly suggest to highlight the pro and cons of each approach in each specific section of the review. I also suggest to comment more in depth which could be the future directions to improve the large animals models use to study cardiac regeneration.

5) Page 8 line 303: provide ref (i.e. Regenerating the human heart: direct reprogramming strategies and their current limitations; DOI: 10.1007/s00395-017-0655-9)

Author Response

Dear Reviewer 3

Thank you for your insightful comments. We had hoped for rigorous feedback from experts in the field to ensure that our review conforms to high academic standards. Thank you for providing such feedback and for taking the time to review our article.

1) We significantly expanded each section with more in-depth information on the various approaches, as well as reporting both the sample size and the most important effect sizes (where possible), to give the reader a better picture of the differences between the approaches and their relative efficacy. Since it was our intention to provide an overview of the entire field of cell-free cardiac regeneration in large animals as well as possible future directions of research, we hope that these changes give a better balance between breadth and depth. The main body of text was expanded from around 4200 to 6200 words.

2) As alluded to in (1) we wished to point toward future directions of research which we found underappreciated or underutilized in large animal models. The concept of direct cardiac reprogramming represents an exciting opportunity for future large animal experiments, which is why we wanted to include it. We have restructured the section to make the reasons for its inclusion in the review more apparent. We have also amended Figure 1 to emphasize direct cardiac regeneration as a future direction in large animal models of cell-free cardiac regeneration.

3) We added a table to each section to summarize the studies presented in the text.

We solicited feedback from a graphic designer and redesigned the Figure accordingly.

4) We moved the discussions of the pros and cons of each method to their respective sections as well as possible future directions of research in that field. We rewrote the conclusion to be more concise and less redundant.

5) We inserted the suggested reference.

Round 2

Reviewer 3 Report

Dear authors,

I really appreciate the extensive work you made to improve your review.

I feel it is now acceptable for publication.